# Investigating the Relationship between Home Parenteral Support and Needs-Based Quality of Life in Patients with Chronic Intestinal Failure: A National Multi-Centre Longitudinal Cohort Study

**DOI:** 10.3390/nu15030622

**Published:** 2023-01-25

**Authors:** Debra Jones, Simon Lal, Chloe French, Anne Marie Sowerbutts, Matthew Gittins, Simon Gabe, Diane Brundrett, Alison Culkin, Chris Calvert, Beth Thompson, Sheldon C. Cooper, Jane Fletcher, Clare Donnellan, Alastair Forbes, Ching Lam, Shellie Radford, Christopher G. Mountford, Daniel Rogers, Rebecca Muggridge, Lisa Sharkey, Penny Neild, Carolyn Wheatley, Philip Stevens, Sorrel Burden

**Affiliations:** 1Division of Nursing, Midwifery and Social Work, School of Health Sciences, University of Manchester, Manchester M13 9PL, UK; 2Intestinal Failure Unit, Salford Royal Foundation Trust, Salford M6 8HD, UK; 3St Mark’s Hospital, London North West University Healthcare NHS Trust, London HA1 3UJ, UK; 4Intestinal Failure and Nutrition Team, Royal Devon and Exeter NHS Foundation Trust, Exeter EX2 5DW, UK; 5GI Medicine, University Hospitals Birmingham NHS Foundation Trust, Birmingham B15 2TH, UK; 6Leeds Gastroenterology Institute, Leeds Teaching Hospitals NHS Trust, Leeds LS9 7JT, UK; 7Norwich Medical School, University of East Anglia, Norwich NR4 7UQ, UK; 8Institute of Clinical Medicine, University of Tartu, 50090 Tartu, Estonia; 9Northern General Hospital, Sheffield Teaching Hospitals NHS Foundation Trust, Sheffield S5 7AU, UK; 10Nottingham University Hospitals NHS Trust, Queens Medical Centre Campus, Nottingham NG7 2UH, UK; 11Newcastle Upon Tyne Hospitals NHS Foundation Trust, Newcastle NE1 4LP, UK; 12Leicester Intestinal Failure Team, Leicester Royal Infirmary, University Hospitals Leicester NHS Trust, Leicester LE1 5WW, UK; 13Gastroenterology, Cambridge University Hospitals NHS Foundation Trust, Cambridge CB2 0QQ, UK; 14Department of Gastroenterology, St. Georges University Hospitals NHS Foundation Trust, London SW17 0QT, UK; 15Patients on Intravenous and Naso-gastric Nutrition Treatment, Christchurch, Dorset BH23 2XS, UK; 16Glasgow Royal Infirmary, Glasgow G4 0SF, UK

**Keywords:** home parenteral nutrition, home parenteral support, quality of life, patient-reported outcomes, longitudinal study, parenteral nutrition impact questionnaire (PNIQ)

## Abstract

Home parenteral support (HPS) is an essential but potentially burdensome treatment that can affect quality of life (QoL). The aims of this longitudinal study were to understand whether any changes in HPS over time were associated with QoL. The Parenteral Nutrition Impact Questionnaire (PNIQ) was used, and data were collected on HPS prescribed at three time points. Data were analysed using multi-level mixed regression models presented as effect size and were adjusted for confounders. Study recruited 572 participants from 15 sites. Of these, 201 and 145 completed surveys at second and third time-points, respectively. PNIQ score was out of 20 with a higher score indicating poorer QoL. Any reduction in HPS infusions per week was associated with an improved PNIQ score of −1.10 (95% CI −2.17, −0.02) unadjusted and −1.34 (95% CI −2.45, −0.24) adjusted. Per day change to the number of infusions per week was associated with a change in the PNIQ score of 0.32 (95% CI −0.15, 0.80) unadjusted and 0.34 (95% CI −0.17, 0.85) adjusted. This is the largest national study to demonstrate improvements in QoL associated with HPS reduction over time using an HPS-specific and patient-centric tool, adding unique data for use of therapies in intestinal failure.

## 1. Introduction

Patients with intestinal failure (IF) have a reduction in gut function to the point where sufficient macronutrients (carbohydrates, protein, and fat), micronutrients, and electrolytes or water can no longer be absorbed by the gut [1]. Home parenteral support (HPS) is a treatment offered to patients with IF in order to sustain life and maintain health [2,3,4].

IF can be classified as type 1 (acute and short term), type 2 (prolonged acute condition, often in metabolically unstable patients), or type 3/chronic (chronic condition, in metabolically stable patients) [5]. Chronic IF (CIF) requires long-term or sometimes life-long HPS, whereby patients receive intravenous infusions for several hours (usually 12–14) for a number of nights every week [6]. Although HPS represents a lifesaving therapy for those with CIF, it is considered a demanding treatment that can lead to serious complications, such as catheter-related bloodstream infections and thrombosis [7,8,9], whereas more long-term adverse complications of CIF include liver failure, osteoporosis, and loss of vascular access [10]. Given that HPS is a time-consuming and restrictive treatment, which often limits the ability to travel, work, and socialise [11,12,13,14,15], it is important to consider the impact this may have on the patient. Indeed, the focus of care for this group of patients should include quality of life (QoL) and consideration for the negative impacts HPS may cause [2,14,16,17]. It is well documented that patients who require HPS have a decreased QoL [2,7,18,19,20,21,22], suffer higher rates of depression and anxiety [11,18,23], and are less likely to ever return to full-time employment [18,24,25].

The most commonly used tools to measure QoL in CIF include Short Form-36 (SF-36) [24,26,27,28], EuroQol-5 Dimension (EQ5D) [27], and Home Parenteral Nutrition-Quality of Life (HPN-QOL) [29,30]. The SF-36 and EQ5D are generic measures of health status and were developed over 30 years ago [31,32,33]. As a result, these tools may now be considered too non-specific to adequately assess QoL in CIF [24]. The HPN-QOL, although specific to CIF, has 57 items and may be difficult to complete in a busy clinical setting in which health care professionals and patients are time poor. The ability to assess patient-reported outcomes for specific treatments is increasingly important to healthcare providers [34,35], such that specific and readily usable QoL measures are now essential for patients with CIF. Other specific QoL tools for HPS include the Short Bowel Syndrome-Quality of Life (SBS-QOL) tool [36]; the Home Parenteral Nutrition Patient-Reported Outcome Questionnaire (HPN-PROQ) [37]; and the Parenteral Nutrition Impact Questionnaire (PNIQ) [38,39]. Although the SBS-QOL and HPN-PROQ offer CIF-specific measurements, the PNIQ was the first HPS-specific tool that was patient centric and measured patient-reported QoL, solely evaluating the impact of parenteral support on an individual’s lived experience. A recent study involving the use of the PNIQ in multiple hospitals in the UK indicated that patients who required more HPS infusions per week reported poorer quality of life [19].

In an effort to gain a deeper understanding of the influence of HPS infusions on patients’ QoL, it is important to know the within-person variation over time. Evidence of the impact of HPS on QoL over time will be useful to inform the use of alternative medical or surgical therapies in this area. This is the first multicentre study to assess QoL longitudinally using a validated, HPS-specific, and patient-centric tool.

Our primary objective was to measure QoL and changes to HPS treatment longitudinally in people with CIF. We hypothesised that changes to a patient’s HPS treatment over time would affect QoL.

## 2. Materials and Methods

The study design was a prospective longitudinal cohort that collected patient survey data over three time-points between March 2020 and March 2022. Surveys were initially sent to participants in March 2020, September 2021, and January 2022. Survey data included patient demographics, clinical HPS characteristics, and QoL scores in patients receiving HPS treatment.

Patients were recruited between 2 March 2020 and 16 December 2021 from 14 NHS hospitals located throughout the UK and from a national patient-led charitable support group for people receiving artificial nutrition: Patients on Intravenous and Nasogastric Nutrition group (PINNT) (15 sites in total). The direct clinical care team at each NHS location and the point of contact at PINNT identified eligible patients and sent them an invitation pack along with the first survey for completion.

Eligible participants included patients aged 18 or over who were living with CIF and receiving HPS (parenteral nutrition or parenteral fluids and electrolytes) in the UK between March 2020 and March 2022. Patients were excluded if they could not give informed consent or could not read or write English.

Participants who consented to study recruitment by completion and return of survey 1 were also asked to consent to further contact by researchers. Those who consented to further contact and provided their contact details were sent two follow-up surveys in September 2021 and January 2022. At each time-point, participants were sent the survey by post or email, with a reminder letter or email being sent three weeks later. For cases in which emails failed or a participant was unable to complete the survey online, a paper version was sent instead if a postal address was available.

The primary outcome was to assess change in QoL over time using the PNIQ measured at three time-points. The PNIQ is made up of 20 questions. The questions are specifically related to HPS and aim to assess QoL and the overall impact of treatment in those who receive HPS. Each statement on PNIQ has a dichotomous response ‘True’ or ‘Not true’. Each response is allocated a score of 1 for ‘True’ and a score of 0 for ‘Not true’. All item scores are summed up to give a total score ranging from 0 (good QoL) to 20 (very poor QoL). Previous validation of the scoring scale of the PNIQ has identified high internal consistency (0.91) and test–retest reliability (0.92) [38].

Data were also collected on demographics (age, gender, ethnicity, etc.), socio-economic characteristics, and clinical HPS characteristics including length of time on HPS, number of infusions per week, changes in care or any surgeries, and the underlying aetiology leading to HPS.

Prior to data collection, we noted that response bias could have influenced the level of QoL reported, as people may be more likely to respond to a survey if they have a better QoL or a better sense of wellbeing on a given day. Equally, people may also be more likely to respond if they are having issues with the topic being asked about (in this case their HPS treatment) and so have a negative view of its impact and how it makes them feel day to day. There may have also been non-response bias if people had difficulty in responding due to ill health or access and ability to use electronic format response on a computer or similar device. To account for these factors, we posted surveys directly to people’s homes or sent the survey via email and allowed several months for responses to be returned. Detection bias could have influenced the association between the receipt of HPS and a reduced QoL if there was a difference in health care and support provided at different hospitals (clustering). A multi-level model was used during analysis to account for clustering within patients and recruitment sites. It was also considered that home support provided by community nurses or family caregivers may affect QoL; however, this has been reported in a cross-sectional study detailing carers’ roles in QoL in patients on HPS [40]. Caregiving was, therefore, not included here as a confounder but is an important factor to consider.

It is also important to note that data for this study were collected during and shortly after the lockdown periods that occurred due to the COVID-19 pandemic. As such, patients’ responses to questions relating to QoL and general well-being may have been negatively impacted [41,42,43]. To account for some of the impact of this, we paused the study during the height of the pandemic (April 2020–August 2020). However, it is of note that a recent cohort study stated that the COVID-19 pandemic had no influence on QoL in a large sample of patients involved in ongoing clinical research, implying that confidence could be placed in patient-reported outcome data collected during this period [44].

The study was approved by the National Research Ethics Committee North West (Ref: 19/LO/1971) and received Research and Development approval. Eligible participants receiving HPS were sent a study pack including an information sheet about the study and contact details of the research team. A receipt of a completed survey was taken as consent.

Patient responses were summarised and reported using standard descriptive statistics. The primary analysis looked to understand the ‘total association’ present as determined from a hypothetical causal relationship between HPS usage (HPS weekly frequency) and the participants’ QoL. The PNIQ scores were assessed for their distributional properties and a multi-level mixed regression model was applied. To account for the multi-level structure of the data and correlation associated within clusters (i.e., responses nested within patients nested within NHS recruitment sites), we used a multi-level mixed model with robust standard errors. The model included random intercept terms associated with patient ID and NHS site. Covariate data were assessed, and whenever possible, appropriate steps were taken to account for the missing values through imputation using the missing by indication method. Initially, a ‘baseline’ model was fitted that included the baseline QoL, time since baseline in months, and the HPS characteristic in instances for which HPS characteristic was a time-varying factor. This baseline model also included the change in the number of infusions per week, which was defined as a categorical change (no change, increase in infusions, and decrease in infusions compared to baseline). We also modelled change in number of infusions as a continuous measure (0 means no change, positive values mean an increase, negative values mean a decrease) to understand if a dose–response relationship was present. These two models were repeated firstly without adjustment, that is, ‘unadjusted’, and with adjustment for pre-defined confounders, that is, ‘adjusted’. The confounders were identified to be age, gender, ethnicity, income, aetiology, type of problem with bowel, if the patient was on antidepressants at any time point, and if the patient had had a recent operation.

## 3. Results

Between March 2020 and March 2022, there were 572 (35%) participants recruited from the 1643 surveys sent out from IF clinics in 14 NHS sites and from PINNT. A participant was considered recruited once they completed and returned a paper-based survey, which was sent to them by a member of their clinical team (survey 1). Of the 572 completed surveys, 382 included ‘signed consent to contact’ from the participant, whereby they agreed to be contacted again by university researchers. Survey 2 was sent in September 2021 (postal and online versions were available), with a reminder sent to those who had not responded after 3 weeks (*n* = 177). Of the 382 participants who completed survey 1, 201 completed survey 2, 128 failed to respond, and 53 were lost to follow up. Survey 3 was sent out in January 2022 by university researchers (postal and online versions were available), with a reminder sent to those who had not responded after 3 weeks (*n* = 45). Of the 201 participants who had completed surveys 1 and 2; 145 completed survey 3, 22 failed to respond, and 34 were lost to follow up. See Figure 1.

### 3.1. Participant Characteristics

Sociodemographic characteristics of participants at baseline are reported in Table 1. Participants at baseline had a mean age of 59.5 (SD 15.9) years, with around two thirds of the participants being female (60.7%), and approximately half were married (52.1%). Despite the majority of participants stating that they had some kind of formal qualification (76.0%), over a quarter said that they were unable to work (29.4%). Approximately half of the participants were retired (49.0%), and so ability to work was not asked or recorded. See Table 1.

### 3.2. Participant Characteristics within Recruitment Sites

Data were gathered on the mean age and percentage gender split for all patients receiving HPS across the different study sites in order to check that the characteristics of the study sample were representative of the total patient population of that site. For 11 of the 15 recruitment sites, the mean age and gender split appeared very similar when comparing the study sample population of the site to the total HPS-dependent population of the site. We could not compare characteristics at two of the sites, as they were unable to provide total population data and so were not included. One site did show a similar gender split between total and study populations; however, the mean age was much lower in the total population. (See Appendix A). Overall, we concluded that the characteristics of the study sample at each site were representative of the total site population, although we cannot rule out selection bias due to other factors such as socio-economic status.

### 3.3. Participant PNIQ Scores and Characteristics of HPS Use, According to Time Point

Missing data meant that PNIQ scores could not be calculated for 20 patients at baseline, one patient at the first follow up, and four patients at the second follow up. For the remaining samples at each time point, the PNIQ score ranged from zero (very good QoL) to 20 (very poor QoL) with mean scores of 11.25 (SD 5.58), 10.68 (SD 5.64), and 11.07 (SD 5.50) at baseline, follow up one, and follow up two, respectively. The mean length of time receiving HPS was 74.02 (range: 1 to 444) months, 84.43 (range: 3 to 441) months, and 97.43 (range: 10 to 447) months at baseline, follow up one, and follow up two, respectively. The mean number of infusions per week (5.52 to 5.69) and number of hours per infusion (12.52 to 12.67) were very similar between time points. The number of patients reporting a change over time to their number of infusions per week was low, with the majority of patients (80–81%) reporting the same number of infusions over the three time points. See Table 2.

### 3.4. Multi-Level Mixed Modelling to Explore the Association between HPS and QoL

A series of multi-level regression models were fitted to better understand the within-person difference in PNIQ score at follow up due to the change, if any, in the amount of HPS infusions per week. As outlined in the Methods section, we fitted four models; models 1 and 2 had the number of HPS infusions set to be categorical and indicated that there was no evidence to suggest that ‘any increase’ in HPS infusions per week at follow up was associated with a change in PNIQ score. However, models 1 and 2 did indicate that for ‘any decrease’ in HPS infusions per week, the PNIQ score changes by −1.10 (95% CI −2.17 to −0.02) and −1.34 (95% CI −2.45 to −0.24) in the unadjusted and adjusted models, respectively. Models 3 and 4 had the number of HPS infusions per week set to be continuous and reported no evidence to suggest the presence of a significant ‘dose response’ change in PNIQ score based on the ‘per day’ change to frequency of infusions per week at follow up. However, these models did demonstrate that a ‘per day’ change in frequency of infusions per week resulted in a change in the PNIQ score by 0.32 (95% CI −0.15 to 0.80) in the unadjusted model and 0.34 (95% CI −0.17 to 0.85) in the adjusted model. This indicates that for each ‘per day’ reduction in infusions per week observed, compared to baseline, the PNIQ score improved by −0.32 (unadjusted) and −0.34 (adjusted). See Table 3.

## 4. Discussion

This is the first study to evaluate QoL over multiple time points using an HPS-specific tool. The aim of this study was to measure PNIQ score longitudinally in patients who were receiving HPS, thus adding to the knowledge base by exploring within-person variation over time. Because previous research demonstrated that patients who required more infusions per week reported a poorer QoL (higher PNIQ score) [19], we hypothesised that within-person changes to HPS treatment over time would also be associated with changes to QoL over time. After adjusting for confounding factors related to the patient’s demographics, we observed that patients with any reduction over time in weekly HPS infusions also reported a decrease in PNIQ score (improved QoL) by as much as 1.34. The ‘per day’ change to the number of infusions per week (continuous change) was associated with an improvement to the PNIQ score of −0.34. This was not influenced by change in the site where the patient was receiving treatment, nor was it influenced by confounding factors such as age, gender, ethnicity, income, underlying disease, type of problem with bowel, if the patient was on antidepressants at any time point, and if the patient had had a recent operation.

A previous smaller study of 20 patients using the generic QoL tool, SF-36, showed that 8 of these patients had a non-statistically significant worse QoL score and an increase in the number of HPS infusions per week compared to baseline [26]. Likewise, a small prospective study that evaluated the effect of an educational booklet on improvement in HPS knowledge in 33 patients concluded that patients who reduced the frequency of HPS infusions had a significant increase in QoL as measured by EQ5D index and physical function domain of the SF-36 [27]. The ability to demonstrate change over time in the QoL in individuals receiving HPS treatment is, of course, relevant to the use of therapies that may reduce HPS requirements [45,46], such as surgical restoration of bowel continuity (bringing colon or distal bowel back into circuit) [47,48,49], intestinal lengthening [50], or treatment with entero-hormones such as glucagon-like peptide-2 analogues [51].

Limitations for this study include the number of participants who had a reduction to the frequency of HPS. Out of the total study population, only 12–13% reported a reduction of infusions over the study period. As the mean length of time receiving HPS at baseline was over 6 years, it is to be expected that this study population may be less likely to see any gut adaptation that would allow for HPS reduction [9,45,52]. However, this factor supports the need for further research in this area, ideally including longitudinal work in more patients following HPS initiation and perhaps specifically evaluating the benefits of the different therapeutic modalities that can help reduce HPS requirements. In addition, data presented were from the UK alone, and 94% of the patients were white. It would be interesting to determine variance between countries for QoL and to include more patients from varying ethnic backgrounds. Data may also be subject to bias due to the use of the missing-indicator method. Although this method is valid for handling missing baseline covariate data, it has been criticized for introducing bias in non-randomized studies [53].

## 5. Conclusions

This is the first and largest national study to evaluate changes in QoL over time using an HPS-specific and patient-centric tool. Patients with CIF who reported a reduction in weekly HPS infusions also reported a decrease in their PNIQ score, indicating improved QoL. The data further demonstrate that the PNIQ tool is sensitive enough to be used in patients receiving HPS and to identify within-person changes over time. Further work will be advantageous in the understanding of how to achieve a reduction in HPS over time and positively influence QoL.

## Figures and Tables

**Figure 1 nutrients-15-00622-f001:**
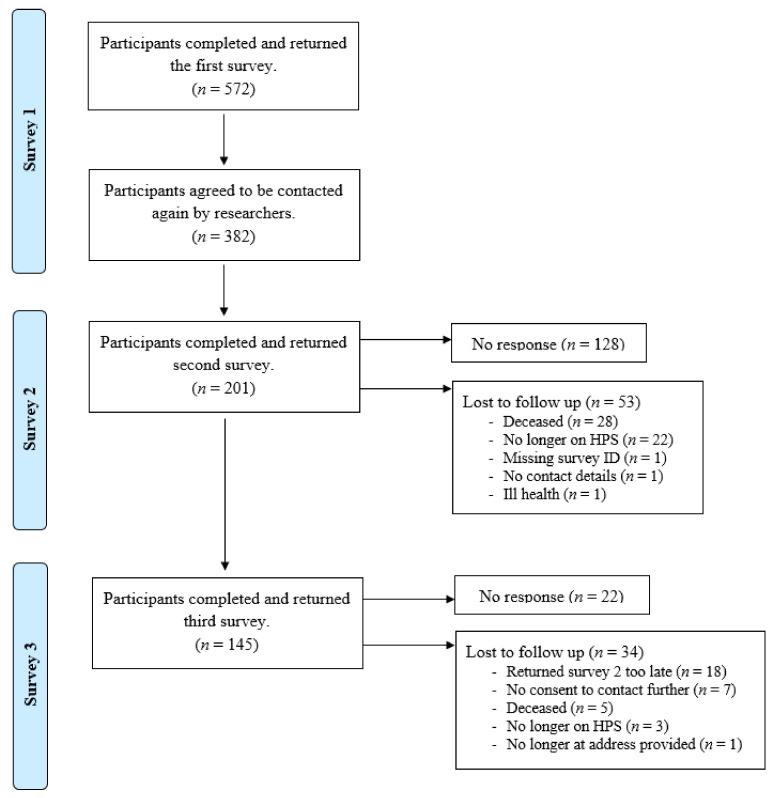
Flow diagram for participant recruitment and retention over time.

**Table 1 nutrients-15-00622-t001:** Sociodemographic characteristics of patients receiving home parenteral support at baseline.

Characteristics	Baseline Patients (N = 572)
Age mean (SD) yrs	59.5 (15.9)
Gender *n* (%)	
Male	219 (38.2)
Female	347 (60.7)
Non-binary/Other/Missing	6 (1.1)
Marital status *n* (%)	
Single	110 (19.2)
Separated/ divorced	64 (11.2)
Living with partner	43 (7.5)
Married	298 (52.1)
Widowed	51 (9.0)
Prefer not to say/missing	6 (1.0)
Occupation *n* (%)	
Employed full time	34 (5.9)
Employed part time	21 (3.7)
Self-employed	10 (1.7)
Homemaker	13 (2.3)
Student	6 (1.0)
Retired	280 (49.0)
Out of work and looking	6 (1.0)
Out of work but not looking	9 (1.6)
Unable to work	168 (29.4)
Other	18 (3.2)
Prefer not to say/missing	7 (1.2)
Highest level of education *n* (%)	
No formal qualifications	104 (18.2)
Trade qualifications	77 (13.4)
GCSE level or equivalent	112 (19.6)
A level or equivalent	55 (9.6)
Higher education diploma	56 (9.8)
Degree/or equivalent	105 (18.4)
Higher degree (MSc/PhD)	30 (5.2)
Prefer not to say/missing	33 (5.8)
Ethnicity *n* (%)	
White	538 (94.0)
Non-White	19 (3.5)
Other/Prefer not say/missing	15 (2.5)
Total income coming into household each month *n* (%)	
Under £250	10 (1.7)
£251–500	31 (5.4)
£501–1000	68 (11.9)
£1001–2000	139 (24.3)
Over £2000	132 (23.1)
Prefer not to say/missing	192 (33.6)
Aetiology	
Inflammatory bowel disease	153 (26.8)
Complications of surgery	123 (21.5)
Radiation damage	14 (2.5)
Ischemia	42 (7.3)
Dysmotility	63 (11.0)
Cancer	42 (7.3)
Other/do not know/missing	135 (23.6)
Type of problem with bowel	
Short bowel	337 (58.9)
Fistula	33 (5.8)
Dysmotility	59 (10.3)
Obstruction	40 (7.0)
Other/do not know/missing	103 (18.0)

N: number for total sample; *n*: number for a particular variable; SD: Standard deviation; yrs: years.

**Table 2 nutrients-15-00622-t002:** Parenteral Nutrition Impact Questionnaire scores and home parenteral support use at baseline, follow up 1, and follow up 2.

	Baseline Patients (N = 572) Mean (SD)	Follow Up 1 Patients (N = 201) Mean (SD)	Follow Up 2 Patients (N = 145) Mean (SD)
QoL outcome variable			
PNIQ score	11.25 (5.58) *n* = 552	10.68 (5.64) *n* = 200	11.07 (5.50) *n* = 141
HPS variables			
Number of months on HPS	74.02 (81.57) *n* = 555	84.43 (83.51) *n* = 199	97.43 (85.20) *n* = 143
Number of Infusions/wk	5.69 (1.52) *n* = 564	5.52 (1.60) *n* = 199	5.57 (1.62) *n* = 144
Hours per infusion	12.58 (2.14) *n* = 563	12.67 (2.27) *n* = 194	12.52 (2.12) *n* = 144
Number of patients with change to infusions/wk			
No change *n* (%)	-	163 (81)	116 (80)
Decrease *n* (%)	-	24 (12)	19 (13)
Increase *n* (%)	-	10 (5)	8 (6)
Missing *n* (%)	-	4 (2)	2 (1)

QoL: quality of life; PNIQ: parenteral nutrition impact questionnaire; HPS: home parenteral support; /wk: per week; N: total number; *n*: number for a particular variable; SD: standard deviation.

**Table 3 nutrients-15-00622-t003:** Multi-level mixed regression analysis to show the impact of change over time in the number of home parenteral infusions per week on PNIQ score (Quality of life).

	Model 1 (*n* = 189) (Unadjusted)	Model 2 (*n* = 184) (Adjusted)	Model 3 (*n* = 189) (Unadjusted)	Model 4 (*n* = 184) (Adjusted)
	Effect Size (95% CI)	Effect Size (95% CI)	Effect Size (95% CI)	Effect Size (95% CI)
Baseline PNIQ Score (independent variable)	0.85 (0.78 to 0.92)	0.88 (0.80 to 0.95)	0.84 (0.77 to 0.91)	0.87 (0.79 to 0.95)
Any increase in HPS Infusions/week compared to no change	−0.59 (−2.08 to 0.89) ^a^	−0.24 (−0.85 to 1.38) ^a^	-	-
Any decrease in HPS Infusions/week compared to no change	−1.10 (−2.17 to −0.02) ^a^	−1.34 (−2.45 to −0.24) ^a^	-	-
Per day change in No. HPS infusions/week	-	-	0.32 (−0.15 to 0.80) ^b^	0.34 (−0.17 to 0.85) ^b^
Time (months)	0.04 (−0.02 to 0.10)	0.05 (−0.01 to 0.12)	0.03 (−0.03 to 0.09)	0.04 (−0.02 to 0.10)
Constant	1.64 (0.51 to 2.76)	1.00 (−1.59 to 3.85)	1.62 (0.49 to 2.75)	0.85 (−1.76 to 3.46)
Variance (NHS site)	1.12 × 10^−13^	6.64 × 10^−19^	3.35 × 10^−13^	5.76 × 10^−19^
Variance (Patient ID)	5.79	4.39	5.83	4.51
Variance (Residual)	3.32	3.41	3.35	3.42

^a^ Categorical change over time in HPS infusions per week (no change = 0; increase = 1; decrease = 2). ^b^ Linear continuous change over time (increase or decrease) in HPS infusions/week. Models 2 and 4 were adjusted for age, gender, ethnicity, income, underlying disease, problem with bowel, if the patient was on antidepressants at any time point, and if the patient had had a recent operation at any time point. PNIQ: parenteral nutrition impact questionnaire; QoL: quality of life; CI: confidence intervals; HPS: home parenteral support.

## Data Availability

The data that support the findings of this study are available on request from the corresponding author.

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
