# Peer review of "Investigating the Relationship between Home Parenteral Support and Needs-Based Quality of Life in Patients with Chronic Intestinal Failure: A National Multi-Centre Longitudinal Cohort Study"

_nutrients, 2023, doi:10.3390/nu15030622_

Round 1
Reviewer 1 Report
The study is well designed and the results well presented. It strength lies in the prospective data collection, multi-center study and relatively large number of patients recruited (572 participants from 15 different sites) allowing for a good cross section of a population with intestinal failure (IF) in the UK.
However, one problem with the study is the large drop out number of participants.
From the initially recruited 572 participants only 201 (35.1 %) completed the QoL questionnaire at the 2nd time point meaning that 64.8 % of participants already dropped out at this point in time. A further 56 participants dropped out when the study was completed with the third questionnaire at the 3rd point in time.
This leaves a total drop out rate of 74.65 %, which will significantly impact on the power of the study and the study hypothesis, that within person change to HPS treatment over time will associated with changes to QoL over time in addition to what is already known (reduction of number of PN infusions per week will improve the QoL of patients with chronic IF).
Whilst participant drop out it is a common problem of questionnaire based surveys the number appears large. The authors should mention in the discussion that the drop out number was large and interpret the results within this context.
The paper makes an important contribution to the field of adult IF management, but requires minor revision.
Author Response
Many thanks for your helpful comments, these have been addressed within the paper and/or responded to below.
Point 1: The study is well designed and the results well presented. It strength lies in the prospective data collection, multi-center study and relatively large number of patients recruited (572 participants from 15 different sites) allowing for a good cross section of a population with intestinal failure (IF) in the UK. However, one problem with the study is the large drop out number of participants. From the initially recruited 572 participants only 201 (35.1 %) completed the QoL questionnaire at the 2nd time point meaning that 64.8 % of participants already dropped out at this point in time. A further 56 participants dropped out when the study was completed with the third questionnaire at the 3rd point in time. This leaves a total drop out rate of 74.65 %, which will significantly impact on the power of the study and the study hypothesis, that within person change to HPS treatment over time will associated with changes to QoL over time in addition to what is already known (reduction of number of PN infusions per week will improve the QoL of patients with chronic IF). Whilst participant drop out it is a common problem of questionnaire based surveys the number appears large. The authors should mention in the discussion that the drop out number was large and interpret the results within this context. The paper makes an important contribution to the field of adult IF management, but requires minor revision.
Response 1: Response bias is discussed in the method section and talks about the population type and why there may be more or less response. We feel that response between surveys may be primarily down to health, given the nature of the treatment. However, this would be speculative and feel that the information given on response bias is adequate enough to cover this point. We hope you agree but please do let us know if you feel more detail should be added.
Reviewer 2 Report
This is a very well written paper and I enjoyed reading it. I have mostly minor points & suggestions.
1. Missing values were dealt with via imputation - this should be discussed under limitations with reflection on how it could impact results
2. Was availability of home support / caregiver considered as a potential confounder? Patients who live alone vs those where a support person who can help with HPN may have different QOL and different response to PS reduction.
3. What happened to the data of the patients who completed the first survey but did not consent to be contacted? Were their baseline data different from the final cohort? Also, how many patients were offered to take the survey ie what was the initial response rate? It would be important to know this for risk of selection bias.
4. There appears to be a high rate of nonresponse from survey 1 to survey 2, but response from survey 2 to survey 3 is relatively high at about 3/4 of those who were still included. This suggests perhaps a more compliant or interested population is completing especially the 3rd survey. Could this impact results?
5. In Figure 1 under survey 1, there is a blue underline from grammar check - please remove this for your final figure
6. 94% of your sample is white - should be mentioned as a limitation. Does this reflect the typical UK HPN patient?
7. It was concluded that the samples were representative of the total HPS dependent population because age/gender were similar. However, this does not rule out selection bias by other factors including socioeconomic etc. This should be discussed in the limitations.
8. Did you account for the possibility that the patients who had reduction in their PS were the ones who were earlier on in their intestinal failure journey ie first few months or <2 years? It is known that QOL can be poor initially when patients start HPN, but over time it tends to improve in many patients. Did you adjust for the possibility that QOL improved over time independently from reduction in PS in those who were new to HPN?
Author Response
Many thanks for your helpful comments, these have been addressed within the paper and/or responded to below.
Point 1: Missing values were dealt with via imputation - this should be discussed under limitations with reflection on how it could impact results
Response 1: Discussion on the imputation method used for dealing with missing data has been added to the limitations section with reflection on impact of results.
Point 2: Was availability of home support / caregiver considered as a potential confounder? Patients who live alone vs those where a support person who can help with HPN may have different QOL and different response to PS reduction.
Response 2: The following has been added to the methods section: ‘It was also considered that home support provided by community nurses or family caregivers may affect QoL, however, this has been reported in a cross sectional study detailing carers’ role in QoL in patients on HPS [40]. Caregiving was therefore not included here as a confounder but is an important factor to consider’. Also see response to point 3 (over analysis of data).
Point 3: What happened to the data of the patients who completed the first survey but did not consent to be contacted? Were their baseline data different from the final cohort? Also, how many patients were offered to take the survey ie what was the initial response rate? It would be important to know this for risk of selection bias.
Response 3: Data from all patients who completed the first survey was kept and analysed together, regardless of consent to contact further. As with point 2 above and point 8 below we were cautious to not create multiple sub groups for multiple analyses which may have resulted in over analysing the data.
Point 4: There appears to be a high rate of nonresponse from survey 1 to survey 2, but response from survey 2 to survey 3 is relatively high at about 3/4 of those who were still included. This suggests perhaps a more compliant or interested population is completing especially the 3rd survey. Could this impact results?
Response 4: Response bias is discussed in the method section and talks about the population type and why there may be more or less response. We feel that response between surveys may be primarily down to health, given the nature of the treatment. However, this would be speculative and feel that the information given on response bias is adequate enough to cover this point. We hope you agree but please do let us know if you feel more detail should be added.
Point 5: In Figure 1 under survey 1, there is a blue underline from grammar check - please remove this for your final figure
Response 5: Thank you for spotting this – blue line has been removed from the figure within the text.
Point 6: 94% of your sample is white - should be mentioned as a limitation. Does this reflect the typical UK HPN patient?
Response 6: The following has been added to the limitations section ‘and 94% of the sample were white. It would be interesting to determine variance between countries for QoL and to include more patients from varying ethnic backgrounds’.
Point 7: It was concluded that the samples were representative of the total HPS dependent population because age/gender were similar. However, this does not rule out selection bias by other factors including socioeconomic etc. This should be discussed in the limitations.
Response 7: We included factors such as socioeconomic status as confounders within the analysis for the total population. The age and gender check was to see if samples for the individual sites appeared to be representative of that site. We have added a sentence to section 3.2 stating ‘, although, we cannot rule out selection bias due to other factors such as socio economic status’.
Point 8: Did you account for the possibility that the patients who had reduction in their PS were the ones who were earlier on in their intestinal failure journey ie first few months or <2 years? It is known that QOL can be poor initially when patients start HPN, but over time it tends to improve in many patients. Did you adjust for the possibility that QOL improved over time independently from reduction in PS in those who were new to HPN?
Response 8: Yes we did consider this during our analysis, however ‘length of time’ was not the primary interest for the study and was not specifically planned for prior to analysis. Also the mean length of time receiving HPS at baseline was 6 years, so already beyond the 2 year point of interest. As we would have had to adjust for ‘length of time’ this would have been an additional exploratory/sensitivity analysis. It may have been interesting to include, but as it was not our focus and as mean length of time was 6 years, we decided against including in order to avoid over analysing the data as with points 2 and 3 above. We did include the following in the discussion: ‘As the mean length of time receiving HPS at baseline was over 6 years, it is to be expected that this study population maybe less likely to see any gut adaptation that would allow for HPS reduction’.